# Enhanced Energy Storage Performance of PVDF-Based Composites Using BN@PDA Sheets and Titania Nanosheets

**DOI:** 10.3390/ma15134370

**Published:** 2022-06-21

**Authors:** Congcong Zhu, Jinghua Yin, Yu Feng, Jialong Li, Yanpeng Li, He Zhao, Dong Yue, Xiaoxu Liu

**Affiliations:** 1School of Materials Science and Chemical Engineering, Harbin University of Science and Technology, Harbin 150080, China; zhucc6@163.com (C.Z.); lypharbin@163.com (Y.L.); yuedong0523@126.com (D.Y.); 2Key Laboratory of Engineering Dielectrics and Its Application, Ministry of Education, Harbin University of Science and Technology, Harbin 150080, China; fengyu@hrbust.edu.cn (Y.F.); zhaohe457296@163.com (H.Z.); 3School of Materials Science and Engineering, Shaanxi University of Science and Technology, Xi’an 710021, China; lijialong@sust.edu.cn

**Keywords:** BN@PDA−STNSs, PVDF-based composites, permittivity, breakdown strength, energy density

## Abstract

With the rapid development of modern electrical and electronic applications, the demand for high-performance film capacitors is becoming increasingly urgent. The energy density of a capacitor is dependent on permittivity and breakdown strength. However, the development of polymer-based composites with both high permittivity (*ε*_r_) and breakdown strength (*E*_b_) remains a huge challenge. In this work, a strategy of doping synergistic dual-fillers with complementary functionalities into polymer is demonstrated, by which high *ε*_r_ and *E*_b_ are obtained simultaneously. Small-sized titania nanosheets (STNSs) with high *ε*_r_ and high-insulating boron nitride sheets coated with polydopamine on the surface (BN@PDA) were introduced into poly(vinylidene fluoride) (PVDF) to prepare a ternary composite. Remarkably, a PVDF-based composite with 1 wt% BN@PDA and 0.5 wt% STNSs (1 wt% PVDF/BN@PDA−STNSs) shows an excellent energy storage performance, including a high *ε*_r_ of ~13.9 at 1 Hz, a superior *E*_b_ of ~440 kV/mm, and a high discharged energy density *U*_e_ of ~12.1 J/cm^3^. Moreover, the simulation results confirm that BN@PDA sheets improve breakdown strength and STNSs boost polarization, which is consistent with the experimental results. This contribution provides a new design paradigm for energy storage dielectrics.

## 1. Introduction

Polymer dielectric capacitors have attracted much attention due to their simple fabrication, excellent mechanical property, high power density, and high breakdown strength. Polymer dielectric capacitors are widely used in electronic and electrical equipment, such as hybrid electric vehicles, power grid systems, aerospace and radar technology, etc. [1,2,3,4]. However, the low energy storage density of polymer dielectric capacitors increases the size and cost of dielectric capacitors and limits their application in modern electric devices. For example, the DC bus capacitors in an electric vehicle power inverter can account for ~35% of the inverter volume and ~23% of the inverter weight [5]. Therefore, polymer dielectrics with high energy storage density are very desirable in developing highly integrated and reliable capacitors.

The discharged energy density *U*_e_ of dielectrics is determined by the applied electric field *E* and dielectric polarization *P* as [6]: $U_{e}=\int_{P_{r}}^{P_{m}}EdP$, where *P*_m_ and *P*_r_ are the maximum polarization and remnant polarization, respectively. Consequently, high *U*_e_ can be obtained by high *P*_m_, low *P*_r_, and large *E*_b_. For linear dielectric materials, the equation can be defined as [7]: $U_{e}=0.5\varepsilon_{0}\varepsilon_{r}E_{b}{}^{2}$, where *ε*_0_ is the vacuum permittivity (8.85 × 10^−12^ F/m), and *ε*_r_ means the relative permittivity. Apparently, dielectrics with high energy density should possess both high *ε*_r_ and *E*_b_. Polymer/ceramic composites exhibit a promising strategy that has the potential of combining the high *E*_b_ of polymer matrix and large *ε*_r_ of the ceramic fillers.

Following the above consideration, in order to improve the polarization and hence *ε*_r_, numerous efforts have been made in preparing 0–3 composites with high filler contents [8,9]. Nevertheless, more interfacial imperfections (such as structural defects, voids, pores, and cracks) and particle agglomeration are also introduced, resulting in increased dielectric loss and decreased *E*_b_. It is well known that large-aspect-ratio one dimensional (1D) and two dimensional (2D) fillers could lead to a higher *ε*_r_ of composites at a lower filler content in comparison with that of the zero dimensional (0D) fillers [10,11,12,13]. Yet, the large difference in *ε*_r_ between ceramic fillers and polymer matrix result in electric field distortion at the interface and decreased *E*_b_.

In order to enhance the *E*_b_ and *U*_e_ of polymer-based composites, tremendous effort has been made [14,15,16]. This study found that adding wide band-gap 2D fillers into the polymer matrix can enhance the *E*_b_ of composites. The 2D fillers can act as conduction barriers to limit the charge migration and increase path tortuosity during breakdown. For example, Zhu et al. introduced the ultra-thin boron nitride nanosheets (BNNS) into a PVDF matrix and a high *E*_b_ of 612 kV/mm and an *U*_e_ of 14.3 J/cm^3^ were obtained [17]. However, the permittivity of wide band-gap fillers is usually low, resulting in the decrease in the permittivity of composites. There is a conflict between high *E*_b_ and *ε*_r_ in the preparation of high-energy storage density composites. In order to solve this problem, many innovative strategies are proposed, including filler modification and optimization, filler orientation distribution, etc., which have greatly improved energy density [18,19,20,21]. Unfortunately, the complex process of filler modification, optimization, and orientation causes a great waste of experimental cost and time and also increases the difficulty of preparation.

Herein, we report a class of solution-processable PVDF-based composites consisting of two 2D fillers with complementary functionalities, including boron nitride sheets coated with polydopamine (BN@PDA) with high *E*_b_ and small-sized titania nanosheets (STNSs) with high permittivity. The PVDF/BN@PDA binary composites without STNSs were prepared and tested for comparison. The results show that the ternary polymer-based composites possess simultaneously increased permittivity and breakdown strength that lead to excellent energy storage performance in comparison with PVDF/BN@PDA binary composites and our previous work [22]. This work provides an easy route for the use of 2D fillers to improve the energy density of composites.

## 2. Materials and Methods

### 2.1. Materials

Boron nitride (h-BN, 99.9% purity) powder was purchased from Shanghai Chaowei Nanotechnology Co., Ltd. (Shanghai, China). Dopamine hydrochloride and tris-(hydroxymethyl)-aminomethane (Tris) were purchased from Aladdin. Poly(vinylidene fluoride) (PVDF) powder with the number-average molecular weight of 534,000 was provided by Sigma-Aldrich Co., Ltd. (Shanghai, China). Potassium carbonate (K_2_CO_3_), lithium carbonate (Li_2_CO_3_), Rutile titania nanoparticles (TiO_2_, diameter of 60 nm), and tetrabutylammonium hydroxide (TBAOH) were purchased from Aladdin Industrial Corporation. The N, N-dimethylformamide (DMF) and anhydrous ethanol were provided by Tianjin Fuyu Fine Co., Ltd. (Tianjing, China).

### 2.2. Preparation of STNSs and Surface Coating of BT by PDA

STNSs were prepared via the top-down exfoliation method [23]. The PDA was coated on the surface of sheets to obtain BN sheets with good dispersion and compatibility. First, 500 mL aqueous solution of Tris was prepared and the pH was adjusted to 8.5; then, 0.8 g dopamine hydrochloride was dispersed into the above solution. Second, the BN sheets were added into the above solution and stirred at 60 °C for 12 h. The obtained BN@PDA sheets were centrifuged and washed with deionized water several times.

### 2.3. Preparation of PVDF/BN@PDA−STNSs Nanocomposites

Pure PVDF, PVDF/BN@PDA, and PVDF/BN@PDA−STNSs composites were prepared via simple physical blending and hot-press method. The 0.5 wt% PVDF/STNSs showed an excellent energy storage performance in our previous work [24]. The appropriate amount of BN@PDA and 0.5 wt% STNSs were sequentially dissolved in 7 mL DMF solvent to form a suspension. The suspension was sonicated to improve fillers’ dispersibility, then mixed with 1 g PVDF powder and stirred for 24 h. The suspension was poured into a clean glass and dried at 80 °C for 10 h. Finally, the samples were hot-pressed at 180 °C for 30 min under 20 MPa pressure. The thickness of the final composite films was approximately 20 ± 5 μm. The schematic illustration of synthesizing the ternary nanocomposites can be seen from Appendix A. In addition, PVDF/BN@PDA binary composites were prepared and tested for comparison.

### 2.4. Characterization

Scanning electron microscopy (SEM, Hitachi SU8020) and the transmission electron microscope (TEM, FEI TECNAI2-12) were used to investigate the surface morphology of BN@PDA and STNSs and the cross-sectional images of the polymer-based composites in this work. The wide-angle X-ray diffraction (XRD) system was performed on X-ray diffractometer (Ultima IV, Rigaku Corporation) using a CuKα source radiation operating at 40 kV and 40 mA. The FTIR spectra of pure PVDF and the composites were tested using JASCO 6100 equipment. At the beam line 1W2A in Beijing Synchrotron Radiation Facility, the small angle x-ray scattering (SAXS) experiments were carried out to research the microstructure of the composites. The storage ring was operated at 2.5 GeV with a current of about 80 mA. The 2D scattering patterns were collected by a Mar165-CCD-charged coupled device and the sample-to-detector distance in the beam direction was 1500 mm. Aluminum electrodes (25 mm in diameter) were deposited on both sides of the samples by vacuum evaporation for subsequent electrical measurements. A broadband impedance analyzer (GmbH Novocontrol Alpha-A) was used to conduct dielectric properties of composites at the frequency range of 10^0^ Hz to 10^6^ Hz. The AC breakdown strength test of composites was carried out via IEC 243. The polarization-electric field (*P-E*) loops of the composites were measured using the Radiant Premier II Ferroelectric Test System at a frequency of 100 Hz.

## 3. Results

### 3.1. Characterization of BN@PDA and STNSs

The morphologies of the BN@PDA and STNSs were characterized by SEM, and the typical images are shown in Figure 1a,b, respectively. The BN@PDA sheets possess a 2D multi-layered structure, and the STNSs present a typical 2D nanosheet structure. Typical TEM images of BN@PDA (Appendix A) reveal that the thickness of PDA was of about 4 nm. The average lateral dimension (d) of BN@PDA as obtained was around 0.5–3 μm and the d of the STNSs was about 0.9–2.8 μm. The thickness of BN@PDA (Appendix A) was about 30–120 nm, and the thickness of the STNSs was about 3 nm, as shown in our previous work [24]. The EDS of BN@PDA is shown in Appendix A, and the presence of O, C, and N elements indicates that the PDA was successfully coated on the BN surface. The EDS of the STNSs is given in Appendix A; it shows the presence of Ti and O elements.

X-ray diffraction (XRD) spectra was used to investigate the crystallization and phase structure of BN@PDA and STNSs (Figure 1e). The peaks at approximately 26.7°, 41.5°, and 55.4° are assigned to the (002), (100) and (111) crystal planes of BN nanosheets, respectively. The characteristic (020) peak at 2*θ* = 9.6° confirms the successful preparation of STNSs. Fourier-transform infrared spectroscopy (FTIR) of BN@PDA and STNSs was carefully examined. As shown in Figure 1f, the peak located at 810 cm^−1^ corresponds to the B−N bending vibration of BN@PDA. The peaks located at 3405 and 1610 cm^−1^ correspond to the −OH stretching vibration and N−H bending vibration of BN@PDA, respectively. As for the STNSs, their characteristic absorption peak of is revealed near ~957 cm^−1^. The −OH stretching vibration and bending vibration peaks appear at 3405 and 1684 cm^−1^, respectively. The −CH_3_ bending vibration and C−N stretching vibration peaks are located at 1370 and 1476 cm^−1^, respectively, which may have been introduced by TBA+ on the STNSs surface.

### 3.2. Characterization of PVDF/BN@PDA−STNSs Composites

The dispersion and morphologies of the BN@PDA in the PVDF composites were characterized using SEM (Appendix A). As in Appendix A, most of the BN@PDA sheets were dispersed homogeneously and well aligned in a certain direction in the matrix when the BN@PDA doping content was 1 wt%. A large amount of sheets aggregated inevitably when the content of BN@PDA was at 10 wt%, as shown in Appendix A. The typical cross-sectional SEM images of the 1 wt% PVDF/BN@PDA−STNSs composite is shown in Figure 2. The BN@PDA and STNSs uniformly dispersed in the PVDF matrix, and no obvious agglomeration and defects can be observed. Abundant hydrogen bonds were formed by rich –OH on the fillers (BN@PDA and STNSs) and fluorine atoms of PVDF chains, which could enhance the dispersal homogeneity of fillers as well as improve the reciprocal interaction between the fillers and PVDF matrix [25,26]. The EDS results (Appendix A) indicate that that BN@PDA and STNSs were introduced into the PVDF matrix.

As illustrated in Figure 3a and Appendix A of the XRD patterns for pure PVDF, PVDF/BN@PDA, and PVDF/BN@PDA−STNSs composites, the characteristic diffraction peaks of inorganic fillers and PVDF matrix were indexed. For example, the (020) plane of the STNSs are well indexed and the characteristic diffraction plane of (020) for BN@PDA sheets is at 2*θ* = 26.7°, respectively. In addition to the characteristic diffraction peaks of inorganic fillers, the characteristic diffraction peaks of the PVDF matrix can also be observed in the composites. The diffraction peaks of 2*θ* = 18.5°and 20.2°are consistent with the (020) plane of *α* phase and (110)(200) plane of *β* phase. Compared with the pure PVDF and PVDF/BN@PDA binary composites, the PVDF/BN@PDA−STNSs composites possess attenuated intensity of an *α* phase peak and enhanced intensity of a *β* phase peak, which indicates that incorporating STNSs is an effective method to obtain a high concentration of polar *β* phase PVDF. The high content of *β* phase PVDF is beneficial in improving the permittivity of the composites [27].

To quantitatively calculate the *β* phase content, FTIR results were obtained on pure PVDF, PVDF/BN@PDA, and PVDF/BN@PDA−STNSs composites, as shown in Appendix A. Compared with pure PVDF, for both series of composites, the hydroxyl group bending vibration and stretching vibration appear at 1684 and 3405 cm^−1^, respectively. The hydrogen bonds generated between the fillers and fluorine atom of PVDF chains via −OH···F-H might form a *β* phase and more dipoles at the interfaces [28]. Figure 3b,c show the enlarged image of FTIR spectra in the wavelength range of 500–1000 cm^−1^. The absorption peaks located at 533 (CF_2_ bonds bending), 614, and 766 (CF_2_ skeletal bending), 796, 855 and 976 cm^−1^ (CH_2_ rocking) correspond to *α* phase PVDF. The peaks centered at 510 (CF_2_ stretching) and 840 cm^−1^ (CH_2_ rocking and CF_2_ stretching) belong to *β* phase PVDF. The *β* phase content (*F(**β)*) in the samples are calculated according to the following equation [29]:(1)$$F(\beta )=\frac{A_{\beta }}{1.26\times A_{\alpha }+A_{\beta }}\times 100\%,$$
where *A*_α_ and *A*_β_ are the absorbance intensities at 766 cm^−1^ and 840 cm^−1^. The *F(β)* of all samples are shown in Figure 3d. Both series of composites possessed increased values of *F(β)*, which indicates that the fillers with rich −OH on the surface could induce *α*-*β* phase transformation [30]. The PVDF/BN@PDA−STNSs ternary composites possessed higher *F(β)* compared with PVDF/BN@PDA composites, which could be ascribed to the presence of more interfaces introduced by STNSs in ternary composites.

The crystalline behavior of all samples was investigated by DSC analysis (Figure 4a and Appendix A). It can be observed that the introduction of the BN@PDA or BN@PDA−STNSs multiphase fillers apparently increased the melting temperature (*T*_m_) of PVDF. The pure PVDF exhibits a *T*_m_ of 162.2 °C, whereas the *T*_m_ values increase to 163.1 and 163.6 °C for the 1 wt% PVDF/BN@PDA and PVDF/BN@PDA−STNSs, respectively. Compared with pure PVDF and binary composites, the ternary composites show higher *T*_m_, which may be ascribed to the enhanced interaction (hydrogen bonding) within the composites. The crystallinity (*X*_c_) was calculated using Equation (2):(2)$$X_{c}=\frac{\Delta H_{m}}{\Delta H_{m-100\%}},$$
where *∆H*_m−100%_ is the melting enthalpy of 100% crystalline PVDF (104.5 J/g). As shown in Figure 4a and Appendix A, the *X*_c_ decreased from 41.8% to 35.6% for PVDF and 35.9% for the composites with 1 wt% BN@PDA and BN@PDA−STNSs, respectively. The decreased *X*_c_ is mainly attributed to the strong interaction at the interface, which can limit the growth of PVDF crystal [31].

Small angle X-ray scattering (SAXS) was used to characterize the microstructure of PVDF, PVDF/BN@PDA, and PVDF/BN@PDA−STNSs composites. Figure 4b and Appendix A show the Porod curves of all samples, and all Porod curves show a negative deviation, which indicates the existence of the interfaces in composites [32,33]. It is well known that the lamellae and amorphous regions are ordered stacked within the spherulites. The long period (*L*_w_) is calculated using Equation (3):(3)$$L_{w}=\frac{2\pi }{q_{\max}},$$
where *q*_max_ is the maximum scattering strength of SAXS profiles. The Lorentz corrected SAXS profiles of PVDF, PVDF/BN@PDA, and PVDF/BN@PDA−STNSs composites are shown in Figure 4c and Appendix A. Notably, the *L*_w_ decreases from 9.37 nm for 1 wt% binary composites to 7.75 nm for the 1 wt% ternary composite. The decreased *L*_w_ value of ternary composite means that the micro-structure of the composites was much denser than that of the PVDF and PVDF/BN@PDA composites. The thickness of the crystalline lamella region (*L*_c_) was calculated using Equation (4):(4)$$L_{c}=L_{w}X_{c},$$
where *X*_c_ is the crystallinity of the sample. The amorphous lamella region (*L*_a_) was calculated using Equation (5):(5)$$L_{a}=L_{w}-L_{c},$$

The *L*_c_ and *L*_a_ of pure PVDF, PVDF/BN@PDA, and PVDF/BN@PDA−STNSs composites are shown in Appendix A. The composites all possessed decreased *L*_c_, which is beneficial in promoting the reversal of dipoles within ferroelectric crystalline domains and thereby decrease the hysteresis loss. Compared with the PVDF/BN@PDA composites, the ternary composites showed a lower value of *L*_c_ (Figure 4d), which may be ascribed to the increased *F(β)* of PVDF/BN@PDA−STNSs composites [34].

### 3.3. Dielectric Properties of the Composites

For dielectric capacitors, high permittivity is one of the crucial characteristic needed for their energy storage performance. The frequency dependence of permittivity (*ε*_r_) and dielectric loss tangent (tan*δ*) of two sets of composites tested at room temperature are shown in Figure 5. As shown in Figure 5a, the introduction of BN@PDA into the PVDF matrix results in decreased permittivity at 100 Hz from 9.2 to 8.0 because of the lower permittivity of BN@PDA (~4) than that of PVDF matrix (~8). Correspondingly, the dielectric loss tangent of PVDF/BN@PDA composites was also reduced (Figure 5b). The PVDF/BN@PDA composites possessed decreased dielectric loss tangent at low frequency, which could be ascribed to the depressed conduction loss and interface polarization loss. With the incorporation of STNSs into PVDF/BN@PDA binary composites, the permittivity of ternary composites increased significantly, as shown in Figure 5c. For example, the ternary composite with 1 wt% BN@PDA exhibited the highest *ε*_r_ of 13.9 at 1 Hz, which is 26.4% higher than that of pure PVDF (~11.0). With the increase in BN@PDA content, the permittivity of ternary composites increases first, and then decreases. Figure 5d shows dielectric loss tangent as a function of frequency for PVDF/BN@PDA−STNSs composites. When the frequency was greater than 3 Hz, the dielectric loss tangent of all ternary composites was lower than that of pure PVDF. The interfacial polarization occurs mainly in the low frequency region due to the long relaxation time. Therefore, the improved permittivity and dielectric loss tangent of the ternary composites are mainly attributed to the enhanced interface polarization. In addition, the increased concentration of electroactive *β* phase *F(β)* and the high permittivity of STNSs might be the reasons for the permittivity enhancement. As the frequency increased further, the dielectric loss tangent of the ternary composites at the same frequency was lower than that of pure PVDF, which may be attributed to the reduction in the crystalline lamella region (*L*_c_).

To further study the effect of BN@PDA and STNSs on the dielectric relaxation of ternary composites, the relaxation imaginary electric modulus (*M*″) of the ternary composites was measured, as shown in Appendix A. The electric modulus (*M**) can be calculated using Equation (6):(6)$$M^{*}=M^{\prime }+iM^{\prime \prime }=\frac{\varepsilon^{\prime }}{\varepsilon^{\prime 2}+\varepsilon^{\prime \prime 2}}+i\frac{\varepsilon^{\prime \prime }}{\varepsilon^{\prime 2}+\varepsilon^{\prime \prime 2}},$$
where *M*″ can be used to analyze the relaxation in dielectric performance without the electrical conduction effect. The interfacial polarization relaxation peak that usually occurs below 1 Hz is not clearly displayed due to the instrument measurement range. Nevertheless, the trough of spectrum shifts towards a higher frequency with the increase in filler content, BN@PDA−STNSs indicating the increasing influence of interfacial polarization. These results are in good agreement with the dielectric properties analysis above.

To study the effect of the STNSs on the conduction of ternary composites, the AC conduction and leakage current density of pure PVDF, PVDF/BN@PDA, and PVDF/composites were tested. Figure 6a,b show the frequency-dependent AC conductivity of two series of composites. The AC conductivity of both sets of composites possessed a strong frequency dependence, suggesting the outstanding insulating performance of the composites. The addition of BN@PDA with excellent insulating performance lead to continuously depressed AC conductivity in PVDF/BN@PDA composites. The introduction of STNSs caused the increased AC conductivity of ternary composites, especially at a low frequency (1 Hz to 10^3^ Hz). The leakage current density of two series of composites is shown in Figure 6c,d. The PVDF/BN@PDA composites possessed significantly decreased leakage current density, which indicates that the BN@PDA sheets have the function of impeding the movement of charges [35]. With the incorporation of the STNSs into PVDF/BN@PDA binary composites, the leakage current density of ternary composites increased. It is worth noting that the leakage current density of ternary composites was lower than that of pure PVDF, which could be attributed to the BN@PDA with high insulating performance. The above results are consistent with the dielectric loss tangent results.

### 3.4. Breakdown and Energy Storage Properties of the Composites

Apart from permittivity, the breakdown strength also plays an important role in improving the energy storage density of composites [36]. The Weibull distribution was used to analyze the breakdown strength of all composites, which is described by Equation (7):(7)$$P(E)=1-\exp[-{\left(\frac{E}{E_{b}}\right)}^{\gamma }],$$
where *P(E)* is the cumulative breakdown probability, *E* means the measured breakdown strength, *E*_b_ is the characteristic breakdown strength, and *γ* is the shape parameter evaluating the scatter of the measured breakdown strength values. A higher *γ* value means a higher level of dielectric reliability [37]. Figure 7a,b show the Weibull distribution of *E*_b_ of PVDF/BN@PDA and PVDF/BN@PDA−STNSs composites. The characteristic breakdown strength of all the samples is summarized in Figure 7c. With BN@PDA contents increasing, the *E*_b_ of the PVDF/BN@PDA composites first increases, and then decreases. The *E*_b_ of PVDF/BN@PDA−STNSs composites possess a similar tendency. Interestingly, the *E*_b_ of the ternary composites introduced by STNSs did not significantly decrease, presenting a value similar to that of the binary composite. For example, the *E*_b_ values of the 1 wt% PVDF/BN@PDA and PVDF/BN@PDA−STNSs composites were 348.8 and 346.7 kV/mm, respectively, which is 24.0% and 23.3% higher than that of pure PVDF (281.2 kV/mm). Based on the above analysis, STNSs were introduced into PVDF/BN@PDA binary composites and significantly improved the permittivity of binary composites, whereas the breakdown strength of binary composites did not deteriorate significantly.

The unipolar polarization-electric field (*P-E*) loops of pure PVDF, PVDF/BN@PDA, and PVDF/BN@PDA−STNSs composites are illustrated in Appendix A. The discharged energy density (*U*_e_) and efficiency (*η*) of pure PVDF and of the composites were calculated from the *P-E* loops and displayed in Figure 7d,e [38]. The *U*_e_ of the binary composite filled with 1 wt% BN@PDA was significantly increased (5.5 J/cm^3^), which could be attributed to the increased *E*_b_ of the binary composites due to the introduction of BN@PDA. Compared with pure PVDF, the *U*_e_ of the binary composite obviously decreased at the same applied electric field because of the decreased permittivity and maximal polarization. The introduction of STNSs caused significantly increased permittivity and maximal polarization of ternary composites, whereas the *E*_b_ did not obviously decrease compared with the binary composites. The PVDF/BN@PDA−STNSs composites possessed improved permittivity and characteristic breakdown strength, yielding the dramatically lifted discharged energy density *U*_e_. In particular, the *U*_e_ of the 1 wt% ternary composite was as high as 12.1 J/cm^3^ at 440 kV/mm, which is 278.1% higher than that of pure PVDF (3.2 J/cm^3^ at 240 kV/mm). The 1 wt% ternary composite showed higher *η* compared with pure PVDF, which is ascribed to suppressed leakage current density.

For comparison, the the *U*_e_ and *E*_b_ of related state-of-the-art composite films filled with different fillers in recent work are summarized in Figure 7f. The samples located below the blue diagonal line always show excellent *U*_e_, and the ones located above the blue diagonal line display high *E*_b_. The expected performance of composite is shown by a blue arrow. The 1 wt% PVDF/BN@PDA−STNSs composite in this work possessed a combination of outstanding *U*_e_ and high *E*_b_, which is superior to the previously reported dielectric materials.

A finite element simulation was performed to investigate the distribution of the electrical potential, electric field, and polarization in the composites. As shown in Figure 8a_1_,a_2_, due to the applied electric field from top to bottom, the electric potential gradually decreased from top to bottom. Due to the large difference of permittivity between the STNSs and PVDF matrix, the isopotential line fluctuated greatly when passing through STNSs. Figure 8b_1_,b_2_ show cross-sectional images of local electrical field distribution for the PVDF/BN@PDA and PVDF/BN@PDA−STNSs composites. The legend with the unit of kV/mm refers to the redistributed local electric field strength in the composites. The BN@PDA sheets with good insulation bore more electric field, which is beneficial in improving the breakdown strength of the composites. STNSs doping brought a small number of weak electric field distortion points, resulting in a slight reduction in the breakdown field strength of the ternary composites compared to that of the binary composites. Figure 8c_1_,c_2_ are the sectional images of the local polarization intensity distribution of the binary and ternary composites, respectively. As can be seen from the figure, in addition to the high polarization intensity region at the interface between the BN@PDA and PVDF matrix, STNSs doping brought a higher interfacial polarization region to the ternary composites. For example, the polarization intensity on the STNSs inside the composite was 1030.5 μC/m^2^.

## 4. Conclusions

In summary, the BN@PDA and STNSs as synergistic dual-fillers in PVDF-based composites have simultaneously enhanced permittivity and breakdown strength. Therefore, the PVDF/BN@PDA−STNSs possess improved discharged energy density. The excellent *U*_e_ was obtained by the elegant combination of STNSs and BN@PDA fillers with complementary functionalities. The increased permittivity of the PVDF/BN@PDA−STNSs composites mainly comes from the STNSs themselves, improved interfacial polarization, and contents of *β* phase PVDF *F(β)*. The significantly enhanced *E*_b_ could be ascribed to the suppressed leakage current density induced by BN@PDA. The method described in this paper demonstrates the advantages of facile preparation and thus provides a general design paradigm for combining complementary structured fillers in polymer composites to improve their comprehensive properties.

## Figures and Tables

**Figure 1 materials-15-04370-f001:**
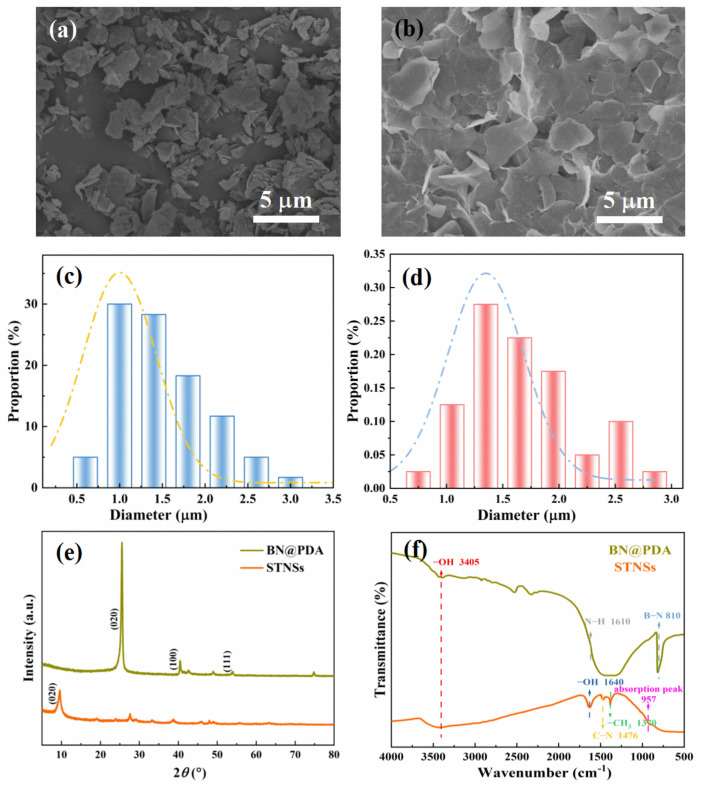
The SEM images of (**a**) BN@PDA and (**b**) STNSs. The diameter distribution of (**c**) BN@PDA and (**d**) STNSs. (**e**) The XRD patterns and (**f**) FTIR spectra of BN@PDA and STNSs.

**Figure 2 materials-15-04370-f002:**
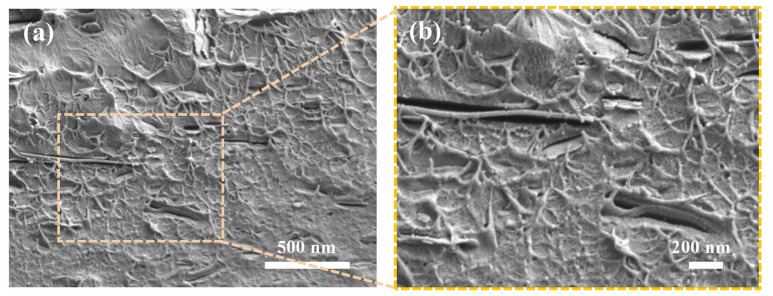
(**a**) The cross-section SEM images of the composites filled with 1 wt% BN@PDA and 0.5 wt% STNSs. (**b**) The enlarged area of (**a**).

**Figure 3 materials-15-04370-f003:**
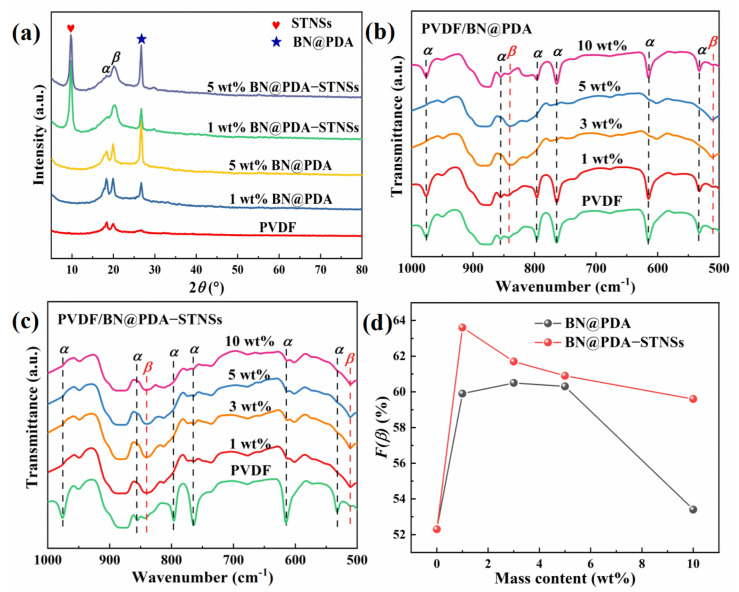
(**a**) The XRD patterns of pure PVDF, PVDF/BN@PDA, and PVDF/BN@PDA−STNSs composites. FTIR spectra of (**b**) the PVDF/BN@PDA and (**c**) PVDF/BN@PDA−STNSs composites. (**d**) *F(β)* of pure PVDF, PVDF/BN@PDA, and PVDF/BN@PDA−STNSs composites.

**Figure 4 materials-15-04370-f004:**
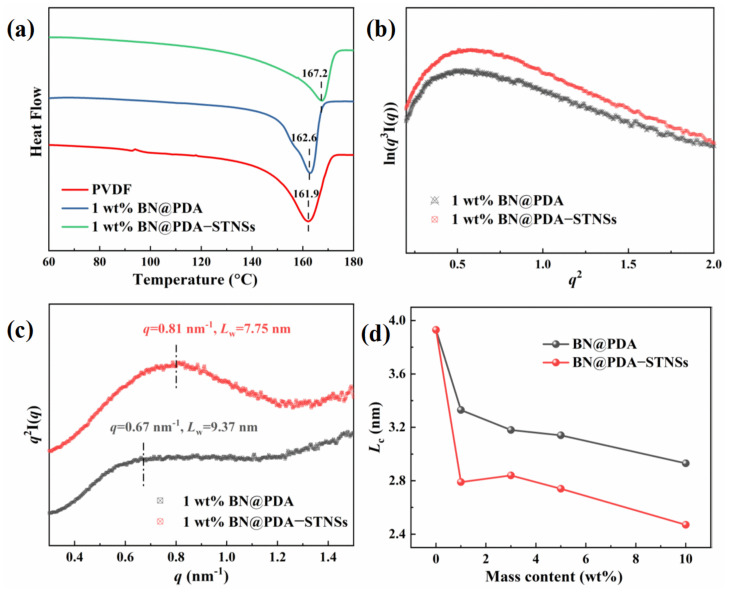
(**a**) The DSC endothermic curves; (**b**) Porod curves; (**c**) Lorentz corrected SAXS profiles; and (**d**) *L*_c_ of pure PVDF, PVDF/BN@PDA, and PVDF/BN@PDA−STNSs composites.

**Figure 5 materials-15-04370-f005:**
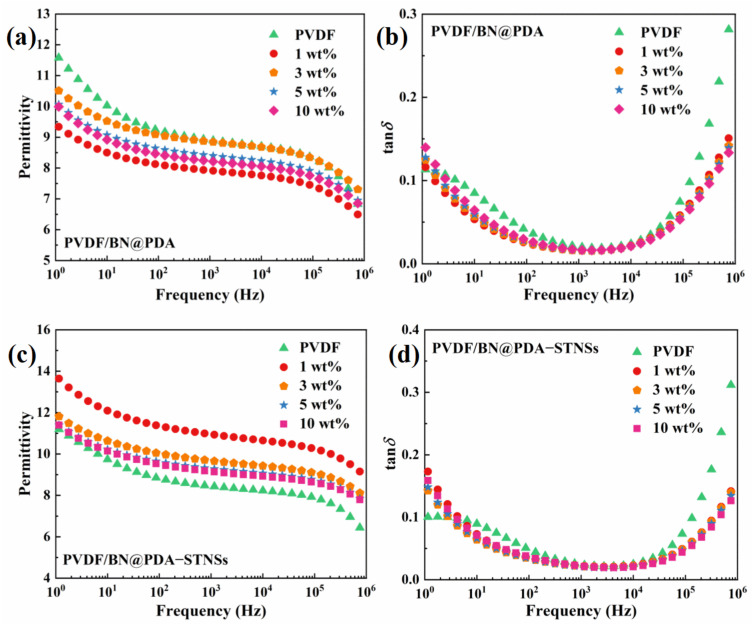
Dielectric permittivity of (**a**) PVDF/BN@PDA and (**c**) PVDF/BN@PDA−STNSs composites. Dielectric loss tangent of (**b**) PVDF/BN@PDA and (**d**) PVDF/BN@PDA−STNSs composites.

**Figure 6 materials-15-04370-f006:**
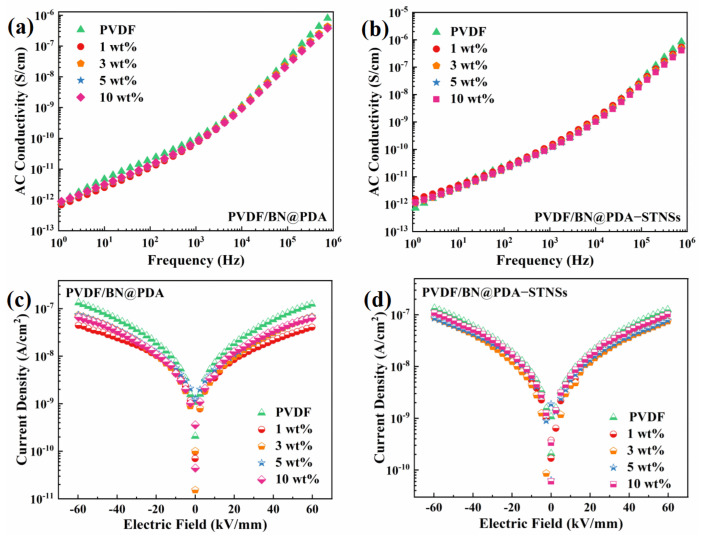
The AC conductivity of (**a**) PVDF/BN@PDA and (**b**) PVDF/BN@PDA−STNSs composites. The current density at 60 kV/mm of (**c**) PVDF/BN@PDA and (**d**) PVDF/BN@PDA−STNSs composites.

**Figure 7 materials-15-04370-f007:**
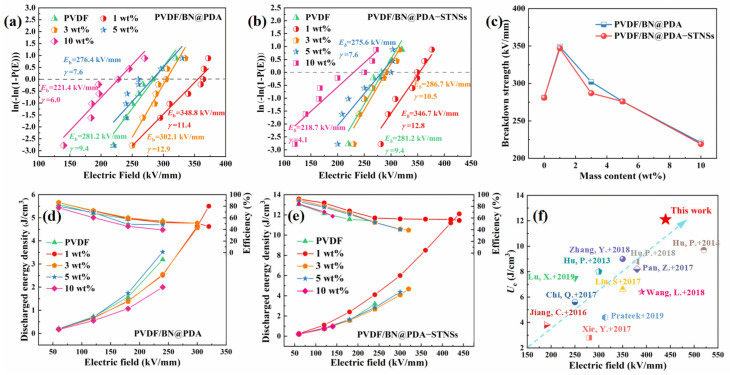
Two-parameter Weibull distribution plots of (**a**) PVDF/BN@PDA and (**b**) PVDF/BN@PDA−STNSs composites. (**c**) Variation of characteristic breakdown strength of PVDF/BN@PDA and PVDF/BN@PDA−STNSs composites with increasing filler content. Discharged energy density (*U*_e_) and efficiency (*η*) of (**d**) PVDF/BN@PDA and (**e**) PVDF/BN@PDA−STNSs. (**f**) Comparison of *U*_e_ and *E*_b_ of related state-of-the-art composites filled with different fillers, respectively [39,40,41,42,43,44,45,46,47,48,49,50].

**Figure 8 materials-15-04370-f008:**
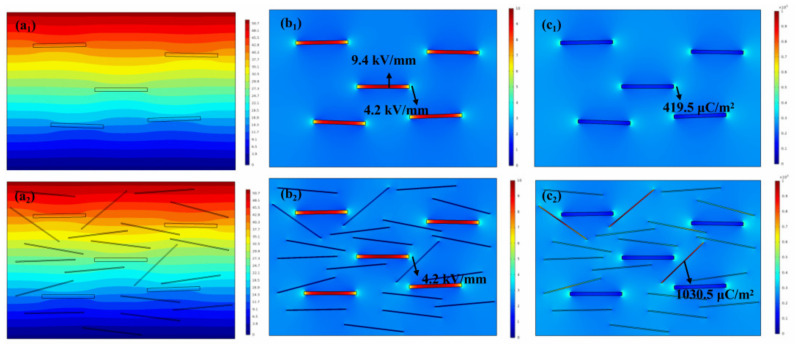
Finite element simulation of electrical potential of (**a_1_**) PVDF/BN@PDA and (**a_2_**) PVDF/BN@PDA−STNSs composites. Finite element simulation of local electric field of (**b_1_**) PVDF/BN@PDA and (**b_2_**) PVDF/BN@PDA−STNSs composites, and local polarization distribution of (**c_1_**) PVDF/BN@PDA and (**c_2_**) PVDF/BN@PDA−STNSs composites.

## Data Availability

Not applicable.

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
