# Peer review of "Enhanced Energy Storage Performance of PVDF-Based Composites Using BN@PDA Sheets and Titania Nanosheets"

_materials, 2022, doi:10.3390/ma15134370_

Round 1
Reviewer 1 Report
The work describes a novel strategy for obtaining high permittivity and breakdown strengths in polymer based composites by using synergistic dual-fillers with complementary functionalities. The fillers used (small-sized titania nanosheets and high insulating boron nitride sheets coated with polydopamine on the surface), introduced into poly(vinylidene fluoride) showed an excellent energy storage performance. This contribution is highly innovative, it advances the state of the art in the field, and clearly provides provide a new design paradigm for energy storage dielectrics. The article is well written and presented, and the results warrant publication in materials, provided the authors address the following minor issues: please add relevant publications for broaden readership DOI: 10.1021/acsami.0c22464; doi: 10.1021/cm0104304; doi: 10.3389/fchem.2020.00058
Author Response
Response: Thanks for the well-meaning proposal, these papers give us great inspiration and we have been cited them in the revised manuscript, see references 3, 4, 11. In addition, the following two documents are also very helpful to us. For citations in the revised manuscript, see references 19, 20.
- Nitti, A.; Po, R.; Bianchi, G.; Pasini, D., Direct Arylation Strategies in the Synthesis of pi-Extended Monomers for Organic Polymeric Solar Cells. Molecules 2016, 22, (1).
- Pasini, D.; Low, E.; Fréchet, J., Novel Design of Carbon‐Rich Polymers for 193 nm Microlithography: Adamantane‐Containing Cyclopolymers. Advanced Materials 2000, 12, (5), 347-351.

Reviewer 2 Report
Authors studied the composites of titania nanosheets and BN sheets in PVDF as energy storage. The authors well characterized structural analysis and energy storage performance depending on the content of fillers. Especially, they found the optimized condition showing an excellent energy storage. Thus, this reviewer suggests the publication in Materials after the minor revision. My comments are below.
- Most figure captions are not explained in details. For example, Figure2a is described as “~ filled with 1 wt%.” No one can know which compound is filled. All figure captions should be modified to get enough information about figures.
- In 3.2, as authors mentioned the homogeneous dispersion of BN-PDA and BN-PDA&STNs, supported by SEM. However, the strategy to prepare homogeneous dispersion is not well described and discussed in the main text. They suggested the hydrogen bond as a mechanism but there is no evidence. At least, some related reference should be mentioned.
Author Response
The authors well characterized structural analysis and energy storage performance depending on the content of fillers. Especially, they found the optimized condition showing an excellent energy storage. Thus, this reviewer suggests the publication in Materials after the minor revision. My comments are below.
Comment 1: Most figure captions are not explained in details. For example, Figure2a is described as “~ filled with 1 wt%.” No one can know which compound is filled. All figure captions should be modified to get enough information about figures.
Response: The reviewer is very professional for our manuscript, and we are really sorry for this mistake. According to your constructive opinion, all figure captions have been modified to get enough information about the figures in the revised manuscript. Related revisions are made as follow:
“Figure 1. The SEM images of (a) BN@PDA and (b) STNSs. The diameter distribution of (c) BN@PDA and (d) STNSs. (e) The XRD patterns and (f) FTIR spectra of BN@PDA and STNSs.
Figure 2. (a) The cross-section SEM images of the composites filled with 1 wt% BN@PDA and 0.5 wt% STNSs. (b) The enlarged area of (a).
Figure 3. (a) The XRD patterns of pure PVDF, PVDF/BN@PDA and PVDF/BN@PDA-STNSs composites. FTIR spectra of (b) the PVDF/BN@PDA and (c) PVDF/BN@PDA-STNSs composites. (d) F(β) of pure PVDF, PVDF/BN@PDA and PVDF/BN@PDA-STNSs composites.
Figure 4. (a) The DSC endothermic curves, (b) Porod curves, (c) Lorentz corrected SAXS profiles and (d) Lc of pure PVDF, PVDF/BN@PDA and PVDF/BN@PDA-STNSs composites.
Figure 5. Dielectric permittivity of (a) PVDF/BN@PDA and (c) PVDF/BN@PDA-STNSs composites. Dielectric loss tangent of (b) PVDF/BN@PDA and (d) PVDF/BN@PDA-STNSs composites.
Figure 6. The AC conductivity of (a) PVDF/BN@PDA and (b) PVDF/BN@PDA-STNSs composites. The current density at 60 kV/mm of (c) PVDF/BN@PDA and (d) PVDF/BN@PDA-STNSs composites.
Figure 7. Two-parameter Weibull distribution plots of (a) PVDF/BN@PDA and (b) PVDF/BN@PDA-STNSs composites. (c) Variation of characteristic breakdown strength of PVDF/BN@PDA and PVDF/BN@PDA-STNSs composites with increasing filler content. Discharged energy density (Ue) and efficiency (η) of (d) PVDF/BN@PDA, (e) PVDF/BN@PDA-STNSs. (f) Comparison of Ue and Eb of related state-of-the-art composites filled with different fillers, respectively[39-50].
Figure 8. Finite element simulation of electrical potential of (a1) PVDF/BN@PDA and (a2) PVDF/BN@PDA-STNSs composites. Finite element simulation of local electric field of (b1) PVDF/BN@PDA and (b2) PVDF/BN@PDA-STNSs composites, and local polarization distribution of (c1) PVDF/BN@PDA and (c2) PVDF/BN@PDA-STNSs composites.”
Comment 2: In 3.2, as authors mentioned the homogeneous dispersion of BN-PDA and BN-PDA&STNs, supported by SEM. However, the strategy to prepare homogeneous dispersion is not well described and discussed in the main text. They suggested the hydrogen bond as a mechanism but there is no evidence. At least, some related reference should be mentioned.
Response: Thanks for your well-meaning comment. We are so sorry that the strategy to prepare homogeneous dispersion is not well described and discussed in the main text. According to your professional opinion, some descriptions have been added in the revised manuscript as follow:
“The appropriate amount of BN@PDA and 0.5 wt% STNSs were sequentially dissolved in 7 ml DMF solvent to form suspension. The suspension was sonicated to improve fillers dispersibility, then mixed with 1 g PVDF powder and stirred for 24 h.”
In addition, two references were cited to prove that the formation of hydrogen bonds at organic-inorganic interfaces can improve the dispersibility of fillers in the matrix in the revised manuscript, see references 25, 26.
(1) Song, Y.; Shen, Y.; Liu, H.; Lin, Y.; Li, M.; Nan, C.-W., Improving the dielectric constants and breakdown strength of polymer composites: effects of the shape of the BaTiO3 nanoinclusions, surface modification and polymer matrix. Journal of Materials Chemistry 2012, 22, (32), 16491.
(2) Xie, Y.; Jiang, W.; Fu, T.; Liu, J.; Zhang, Z.; Wang, S., Achieving High Energy Density and Low Loss in PVDF/BST Nanodielectrics with Enhanced Structural Homogeneity. ACS Applied Materials Interfaces 2018, 10, (34), 29038-29047.

Reviewer 3 Report
Following are some of the concerns that need to be addressed:
1. The authors should clearly mention the novelty, especially when compared with their previous paper(Zhu, C., Yin, J., Li, J., Li, Y., Zhao, H., Yue, D., ... & Liu, X. (2021). Enhanced energy storage of polyvinylidene fluoride‐based nanocomposites induced by high aspect ratio titania nanosheets. Journal of Applied Polymer Science, 138(16), 50244.).
2. There are several phrases used in this manuscript which are similar to others. The authors should work on minimizing the similarity.
3. A thorough proofreading of the manuscript is required as the quality of the language, such as grammar, syntax, spelling, and readability of the manuscript in English, needs major improvement.
Author Response
Following are some of the concerns that need to be addressed:
Comment 1: The authors should clearly mention the novelty, especially when compared with their previous paper(Zhu, C., Yin, J., Li, J., Li, Y., Zhao, H., Yue, D., ... & Liu, X. (2021). Enhanced energy storage of polyvinylidene fluoride‐based nanocomposites induced by high aspect ratio titania nanosheets. Journal of Applied Polymer Science, 138(16), 50244.).
Response: The reviewer is very careful and responsible, thank you for giving us the opportunity to modify our manuscript. Indeed, compared with the Ue of our previous paper in 2021 (~0.32 J/cm3), the excellent Ue (~12.1 J/cm3) in this work is obtained by the elegant combination of STNSs and BN@PDA fillers with complementary functionalities. The fillers in our previous paper in 2021 are titania nanosheets (TNSs). The average size of most TNSs is in a range of 25-45 μm. The lateral size of titania nanosheets was reduced by controlling calcination time. The lateral size of small titania nanosheets (STNSs) is about 0.9-2.8 μm. In our previous paper in 2022 [1], we demonstrate a facile and highly efficient approach, namely, adjusting the size of titania nanosheets (TNSs) as a two-dimensional (2D) filler to dramatically enhance the Ue (~11.7 J/cm3) of polymer dielectrics by simultaneously adjusting the energy gap and enhancing the interface effect of the 2D nanosheets. In this work, we report a class of solution-processable PVDF-based composite consisting of two 2D fillers with complementary functionalities, including boron nitride sheets coated with polydopamine (BN@PDA) with high Eb and STNSs with high permittivity. The preparation approach of synergistic dual-filler composites in this work can not only achieve high Ue, but also has the advantage of facile preparation. According to your constructive opinion, we have supplemented the novelty of this work compared to our previous paper in 2021. Related revisions are made as follow:
“The results show that the ternary polymer-based composites possess simultaneously increased permittivity and breakdown strength, which lead to excellent energy storage performance in comparison with PVDF/BN@PDA binary composites and our previous work[22]. ”.
Comment 2: There are several phrases used in this manuscript which are similar to others. The authors should work on minimizing the similarity.
Response: Thank you very much for your valuable comments and we fully agree with your comments. According to your comments, we have reduced the similarity in phrase usage in this manuscript. For example, we have modified “Compared with pure PVDF, the Ue of the binary composite is obviously decreased at the same applied electric field due to the decreased permittivity and maximal polarization. ” into “Compared with pure PVDF, the Ue of the binary composite is obviously decreased at the same applied electric field because of the decreased permittivity and maximal polarization. ”
we also have modified “Apart from the characteristic diffraction peaks of inorganic fillers, the characteristic diffraction peaks of PVDF matrix can also be observed in the composites. ” into “In addition to the characteristic diffraction peaks of inorganic fillers, the characteristic diffraction peaks of PVDF matrix can also be observed in the composites. ”
Comment 3: A thorough proofreading of the manuscript is required as the quality of the language, such as grammar, syntax, spelling, and readability of the manuscript in English, needs major improvement.
Response: Thanks for your well-meaning comment. Your comment is very professional and constructive. We have thoroughly proofread the manuscript and improved the language quality, such as grammar, syntax, spelling and readability of the English manuscript. For example, we have modified “The AC conductivity of both sets of composites all possess strong frequency dependence, suggesting the outstanding insulating performance of the composites.” into “The AC conductivity of both sets of composites all possess strong frequency dependence, suggesting the outstanding insulating performance of the composites.”.
We also have removed “This section may be divided by subheadings. It should provide a concise and precise description of the experimental results, their interpretation, as well as the experimental conclusions that can be drawn.”

Round 2
Reviewer 3 Report
The authors have incorporated the comments in the revised manuscript.